# *Guettarda crispiflora* Vahl Methanol Extract Ameliorates Acute Lung Injury and Gastritis by Suppressing Src Phosphorylation

**DOI:** 10.3390/plants11243560

**Published:** 2022-12-16

**Authors:** Dahae Lee, Ji Won Kim, Chae Young Lee, Jieun Oh, So Hyun Hwang, Minkyeong Jo, Seung A Kim, Wooram Choi, Jin Kyoung Noh, Dong-Keun Yi, Minkyung Song, Han Gyung Kim, Jae Youl Cho

**Affiliations:** 1Department of Integrative Biotechnology, Sungkyunkwan University, Suwon 16419, Republic of Korea; 2Instituto de BioEconomia, Quito 170135, Ecuador; 3International Biological Material Research Center, Korea Research Institute of Bioscience and Biotechnology, Daejeon 34141, Republic of Korea; 4Research Institute of Biomolecule Control, Sungkyunkwan University, Suwon 16419, Republic of Korea; 5Biomedical Institute for Convergence at SKKU, Sungkyunkwan University, Suwon 16419, Republic of Korea

**Keywords:** *Guettarda crispiflora* Vahl, inflammation, Src/NF-κB pathway, gastritis, acute lung injury

## Abstract

Many species in the genus *Guettarda* are known to exert anti-inflammatory effects and are used as traditional medicinal plants to treat various inflammatory symptoms. However, no studies on the inflammatory activities of *Guettarda crispiflora* Vahl have been reported. The aim of the study was to investigate in vitro and in vivo the anti-inflammatory effects of a methanol extract of *Guettarda crispiflora* Vahl (Gc-ME). To determine the anti-inflammatory activity of Gc-ME, lipopolysaccharide (LPS)-, poly(I:C)-, or Pam3CSK4-treated RAW264.7 cells, HCl/EtOH- and LPS-treated mice were employed for in vitro and in vivo tests. LPS-induced nitric oxide production in RAW264.7 cells was determined by Griess assays and cytokine gene expression in LPS-activated RAW264.7 cells, confirmed by RT- and real-time PCR. Transcriptional activation was evaluated by luciferase reporter gene assay. Target protein validation was assessed by Western blot analysis and cellular thermal shift assays (CETSA) with LPS-treated RAW264.7 and gene-transfected HEK293 cells. Using both a HCl/EtOH-induced gastritis model and an LPS-induced lung injury model, inflammatory states were checked by scoring or evaluating gastric lesions, lung edema, and lung histology. Phytochemical fingerprinting of Gc-ME was observed by using liquid chromatography–mass spectrometry. Nitric oxide production induced by LPS and Pam3CSK4 in RAW264.7 cells was revealed to be reduced by Gc-ME. The LPS-induced upregulation of iNOS, COX-2, IL-6, and IL-1β was also suppressed by Gc-ME treatment. Gc-ME downregulated the promotor activities of AP-1 and NF-κB triggered by MyD88- and TRIF induction. Upstream signaling proteins for NF-κB activation, namely, p-p50, p-p65, p-IκBα, and p-Src were all downregulated by Ch-EE. Moreover, Src was revealed to be directly targeted by Gc-ME. This extract, orally treated strongly, attenuated the inflammatory symptoms in HCl/EtOH-treated stomachs and LPS-treated lungs. Therefore, these results strongly imply that *Guettarda crispiflora* can be developed as a promising anti-inflammatory remedy with Src-suppressive properties.

## 1. Introduction

Inflammation is an innate immune response triggered by various exogenous and endogenous stimuli [1,2]. Inflammation is mediated by cells of the innate immune system, such as macrophages and neutrophils, which secrete various cytokines and chemokines [3,4]. Pattern recognition receptors (PRRs; e.g., Toll-like and nod-like receptors) are expressed on cells of the innate immune system and can recognize a variety of pathogen-associated molecular patterns (PAMPs) of bacteria, viruses, and fungi. Through the binding of PAMPs to PRR, pathogenic signals are transduced into the cells of the innate immune system, and inflammation can be initiated [5]. One of the types of PRR, Toll-like receptors (TLRs), can bind to a variety of pathogenic ligands. For instance, TLR4 binds to lipopolysaccharides (cell wall components of Gram-negative bacteria), TLR3 binds to polyinosinic:polycytidylic acid [poly(I:C)], and TLR1/2 binds to pam3CysSerLys4 (Pam3CSK4) [6,7]. Adaptor molecules transduce TLR-ligand signals [8]. TRIF and MyD88 are the most upstream adaptor molecules initiating the sequential kinase cascade in which Src, AKT, IKK, IKBα, and p50/p65 (NF-κB) are phosphorylated to activate transcription factors for inflammatory cytokines and enzymes [9,10,11]. The activation and nuclear localization of NF-κB induce the transcription of pro-IL1β, iNOS (inducible nitric oxide synthase), cyclooxygenase-2 (COX-2; which induces prostaglandin synthesis), and interleukin-6 (IL-6), which promote and initiate inflammation [12].

While inflammation is a critical protective mechanism for the body, prolonged and severe inflammation is associated with many diseases. For instance, gastritis is the chronic inflammation of the lining of the stomach, characterized by stomach ulcers and the elevation of inflammatory markers [13]. Acute lung injury (ALI) is a form of acute respiratory failure characterized by severe inflammation, which can develop into acute respiratory distress syndrome (ARDS), a life-threatening lung disease [14]. 

Herbal medicines have been widely used to treat inflammatory diseases. Many species in the genus *Guettarda* are known to exert anti-inflammatory effects and are used as traditional medicinal plants to treat various inflammatory symptoms. *Guettarda speciosa* L. inactivates Syk and JNK which are involved in the production of inflammatory mediators [15]. *Guettarda speciose* L. also inhibits cyclo-oxygenase-1 (COX-1) activity and ameliorates acute lung injury [16,17]. In light of these findings, we, for the first time, examined the anti-inflammatory activity of *Guettarda crispiflora* Vahl methanol extract (GC-ME).

## 2. Results

### 2.1. Effects of Gc-ME on NO Protection

As we began the study, we first assessed the degree of inhibition of nitric oxide synthesis by Gc-ME. Nitric oxide (NO) is a signaling molecule that plays an important role in inflammation, and NO production is correlated with the degree of inflammation [18]. To confirm this, Gc-ME was added to RAW264.7 cells, a mouse-derived macrophage cell line, and NO production decreased in a concentration-dependent manner (Figure 1A). To trigger an inflammatory response via TLRs, we treated RAW264.7 cells with LPS (a TLR4 ligand), Poly (I:C), a TLR3 ligand, and pam3CSK4, a TLR1/2 ligand [19]. The inhibition of NO production by Gc-ME was not statistically significant when the RAW264.7 cells were co-incubated with Poly(I: C) (Figure 1B), implying that Gc-ME might not be effective in virus-induced inflammatory symptoms. However, it was confirmed that the synthesis of NO triggered by Pam3CSK4, a TLR3 ligand, treatment, was reduced by this extract (Figure 1C). In addition, Gc-ME did not produce cytotoxicity in RAW264.7 cells at any concentration tested. (Figure 1D).

We selected prednisolone, a drug commonly used to control inflammation, as a positive control when evaluating the anti-inflammatory efficacy of Gc-ME. Prednisolone dose-dependently reduced NO production in RAW264.7 cells triggered by LPS without displaying cytotoxicity (Figure 1E,F) at a similar level as reported previously [20,21].

Additionally, LC-MS/MS analysis was conducted to characterize the phytochemicals contained in Gc-ME (Figure 1G): 64 components were identified, including divaricatol and odoratin-7-O-β-D-glucoside at a retention time (RT) of 1.05 min; baohuoside I at an RT of 3.57 min; 2′,6′-dihydroxy-4,4′-dimethoxydihydrochalcone at an RT of 6.51 min; and apigenin-7-O-β-D-glucurono-pyranoside at an RT of 15.20 min, as shown in Figure 1G and Appendix A. Of particular interest, divaricatol has been reported to have an analgesic effect, and baohuoside I and 2′,6′-dihydroxy-4,4′-dimethoxydihydrochalcone have anti-inflammatory effects [22,23,24,25].

### 2.2. Effects of Gc-ME on Inflammatory Cytokine mRNA Expression

We assessed cytokine mRNA expression, which increases during inflammation. First, a representative pro-inflammatory cytokine (e.g., IL-1) was selected, and the expression of the corresponding mRNA was confirmed through RT-PCR [26,27]. After confirming the expression of inflammatory cytokine mRNA in general, the expression of the cytokines of interest—IL-1β, iNOS, COX-2, and IL-6—was assessed following LPS stimulation; all were confirmed to increase. The expression of all the assessed cytokine mRNAs in the LPS-stimulated cells was reduced in a concentration-dependent manner when co-incubated with Gc-ME (Figure 2A). In addition, a similar inhibitory pattern of these cytokines was also confirmed by real-time PCR (qRT-PCR) analysis [Appendix A].

Next, we tested whether or not the activities of transcription factors having effects on the expression of these mRNAs were regulated by Gc-ME after testing its viability (Figure 2B). The activation of the transcription factors in the inflammatory signaling pathways was induced by the transfection of the TRIF and MyD88 vectors (Figure 2C–F). The activity of each transcription factor was reduced in a concentration-dependent manner in the transfected cells incubated with Gc-ME (Figure 2C–F). Both AP-1 and NF-κB have important roles as key transcription factors that modulate the expression of many genes through complex intracellular signal transduction cascades. The activities of both of these transcription factors were inhibited by Gc-ME, another indicator of the anti-inflammatory effects of Gc-ME (Figure 2C–F).

Next, we focused on NF-κB to identify the mechanism and molecular targets of inflammation suppression by Gc-ME. Therefore, the expression and activation of the sub-signal transmitting substances of the corresponding signaling pathway were confirmed (Figure 2G).

### 2.3. Effects of Gc-ME on Protein Activation

Next, to determine the target(s) of Gc-ME, we assessed other factors belonging to inflammation signaling pathways. After pre-treatment with Gc-ME, LPS was added for 5 to 60 min, and the phosphorylation of IκBα was confirmed. The phosphorylation of IκBα was reduced at all induction time points except 15 min, showing a disappearance of the IκBα protein as a consequence of the strong phosphorylation of IκBα at 5 min and its subsequent proteolytic degradation (Figure 3A), as reported previously [28,29,30]. The total protein level of IκBα was found to be recovered at 5 and 15 min, while it was not increased at 30 and 60 min (Figure 3A), implying that the inhibition of IκBα activation signaling can appear at early time points.

When the expression and activation of Src, which is an upstream signal-transmitting substance, were controlled, the phosphorylation of Src was suppressed in all cases of inflammatory induction. The results indicated that the anti-inflammatory effects of Gc-ME are due to the inhibition of Src protein activation (Figure 3B). We also used a wild type of Src and its SH2- and SH3-domain deletion mutants to identify the specific domain of Src to which Gc-ME binds (Figure 3C–E). The phosphorylation of all mutant Src proteins was suppressed, indicating that neither the SH2 nor SH3 domain is a target of Gc-ME. In addition, Gc-ME was likely bound to SH1, another kinase domain of Src (Figure 3C–E). The results were confirmed in cells overexpressing Src. We proceeded with the cellular thermal shift assay (CETSA) and confirmed that the thermal stability of Src was increased through interaction with Gc-ME (Figure 3F).

### 2.4. Anti-Inflammatory Effects of Gc-ME in an HCl/EtOH-Induced Mouse Model of Gastritis

To verify the anti-inflammatory effect of Gc-ME in vivo, we established an oral HCl/EtOH-induced mouse model of gastritis. Inflammatory lesions in the stomach were reduced in a dose-dependent manner among mice in the 50 and 100 mg/kg Gc-ME groups in (Figure 4A and Appendix A). These effects were comparable to those of the anti-ulcer drug, ranitidine. At the transcription level, the expression of COX-2, IL-1β, and iNOS was significantly reduced in Gc-ME-treated mice (Figure 4B and Appendix A). Additionally, as shown in Western blotting, Gc-ME treatment reduced the phosphorylation of Src and IκBα, with the 100 mg/kg Gc-ME group showing effects comparable to ranitidine (Figure 4C).

### 2.5. Gc-ME Ameliorates Acute Lung Injury in an LPS-Induced ALI Model

To investigate the anti-inflammatory effects of Gc-ME in vivo, we used a mouse model of ALI induced by the intranasal administration of LPS. In Gc-ME-treated mouse groups, the wet-weight: dry-weight ratio of the lung was reduced compared with the LPS-only group; these findings indicate that pulmonary edema caused by inflammation was alleviated in the Gc-ME group (Figure 5A). Similar to the results of the in vitro assays, the expression of iNOS was significantly reduced following Gc-ME administration (Figure 5B and Appendix A). In addition, Gc-ME attenuated the phosphorylation of Src, which we considered a molecular target of Gc-ME (Figure 5C). The ALI score was measured in H&E-stained lung sections (Figure 5D). In the Gc-ME-treated group, multiple parameters of lung injury, including neutrophil recruitment, hyaline membrane formation, proteinaceous debris in airspaces, and alveolar septal thickening, were significantly reduced (Figure 5E).

## 3. Discussion

Inflammation is triggered by various pathogenic stimuli and tissue damage [31]. When all the inciting antigens and possible triggers disappear, inflammation is terminated naturally by the innate immune system and cytokine networks promoting wound healing and tissue repair [32,33]. However, the perturbation of acute inflammation in a short time period or uncontrolled, chronic inflammation leads to tissue pathology and disease [4,5,32,34]. Thus, therapeutic agents controlling pathological inflammation and targeting molecular mediators of inflammation are crucial [35,36]. 

*G. crispiflora* Vahl is a plant belonging to the genus *Guettarda* Linn, and its native range is tropical America. Based on previous reports indicating that genus *Guettarda* inhibited inflammatory mediators [37,38], we investigated the anti-inflammatory effects of *G. crispiflora* Vahl in this study. Nitric oxide is an inflammatory mediator and is produced upon the activation of macrophages [39]. Thus, we examined the NO inhibition activity of Gc-ME using LPS-, poly- (I:C)-, or Pam3CSK4-induced macrophages. In LPS- and Pam3CSK4-induced macrophages, Gc-ME reduced NO production in a concentration-dependent manner compared with the positive control. We confirmed the lack of cytotoxicity of 50, 75, or 100 μg/mL Gc-ME in both RAW264.7 and HEK293T cells. Further, we conducted RT-PCR to assess the inhibition of inflammatory cytokines and enzymes at the transcription level and found that Gc-ME reduced the expression of iNOS, COX-2, IL-1β, and IL-6 mRNAs in a concentration-dependent manner. In addition, the overexpression of Flag-MyD88 and CFP-TRIF in a luciferase assay confirmed that the inhibition of the transcription of inflammatory cytokine and enzymes is mediated by transcription factor NF-κB. The p50/p65 subunits of NF-κB are bound to IκBα, phosphorylated by IKK, and released upon the degradation of IκBα [29,30]. To find the upstream molecular target of Gc-ME, we conducted Western blotting and showed that the phosphorylation of p50, p60, IκBα, AKT, and Src is reduced by Gc-ME. Therefore, we confirmed that NO synthesis, the expression of mRNA for inflammatory genes, and a key kinase cascade were suppressed by Gc-ME treatment. These findings suggest that a possible molecular target of Gc-ME is Src, the initial non-receptor tyrosine kinase controlling multiple signaling pathways. To verify whether Gc-ME interacts with and inhibits Src activity, the overexpression of SH2- or SH3-deleted Src and CETSA was analyzed. The enhanced stability of *p*-Src in Gc-ME-treated samples revealed that Gc-ME interacts with Src, and the domain deletion assay showed that the kinase domain (SH1) is the binding site. The LC/MS data proved that Gc-ME contains divaricatol, baohuoside 1, and apigenin-7-O-β-D-glucuronopraoside, previously reported to exert anti-inflammatory activity [40]. Taken together, these results indicate that Gc-ME exerts anti-inflammatory effects in vitro and that its molecular target is Src. 

To examine the anti-inflammatory activity of Gc-ME in vivo, mouse models of gastritis and ALI were established [41,42]. In the mouse gastritis model, Gc-ME administration ameliorated stomach ulcers and reduced the expression of inflammatory genes (COX-2, IL-1β, IL6, and iNOS) and the phosphorylation of Src and IκBα. This result supports that Gc-ME alleviated gastritis by suppressing Src activity. In the mouse ALI model, pulmonary edema and acute lung injury score (as identified by H&E staining) were significantly decreased by oral administration of Gc-ME. Additionally, RT-PCR and Western blot data once more confirmed the anti-inflammatory effect of Gc-ME and its molecular target. Together, the in vivo results indicate that Gc-ME suppresses the NF-κB pathway by binding Src and can reduce the severity of gastritis and acute lung injury. The occurrence of gastritis and ALI has been increasing worldwide, and therapeutic agents without significant safety concerns are needed. For instance, non-steroidal anti-inflammatory drugs (NSAIDs) could worsen pre-existing heart failure or induce significant cardiovascular events, hypertension, and intestinal ulcers [43,44]. Thus, we suggest Gc-ME as a possible therapeutic agent for treating gastritis and acute lung injury. During the study, we observed an antioxidant effect of Gc-ME by ABTS assay and DPPH assay. The radical scavenging effects of Gc-ME will be investigated in the future.

## 4. Materials and Methods

### 4.1. Materials and Reagents

A 95% methanol extract of *Guettarda crispiflora* (Gc-ME) was obtained from the Plant Extract Bank of the Plant Diversity Research Center, Daejeon, Korea (FBM 160-095, Ecuador). We purchased pam3CSK4 (Pam3CysSerLys4), poly(I:C), lipopolysaccharide (LPS), (3-4,5-dimethylthiazol-2-yl)-2,5-diphenyl-tetrazolium bromide (MTT), polyethylenimine (PEI), prednisolone, and dexamethasone from Sigma Chemical Co. (St. Louis, MO, USA). The RAW264.7 and HEK293T cell lines were purchased from the American Type Culture Collection (ATCC; Rockville, MD, USA). Fetal bovine serum (FBS), phosphate-buffered saline (PBS), Roswell Park Memorial Institute (RPMI) 1640 medium, penicillin, streptomycin, and Dulbecco’s modified Eagle’s medium (DMEM) were purchased from GIBCO (Grand Island, NY, USA). Antibodies to native or phosphorylated p50, p65, IκBα, and HA (hyaluronic acid) were purchased from Cell Signaling Technology (Beverly, MA, USA), and antibodies to native or phosphorylated Src and β-actin were obtained from Santa Cruz (Dallas, TX, USA).

### 4.2. Preparation of G. crispiflora Vahl Extract and Its Use

*Guettarda crispiflora* Vahl originally was collected in the San Juan de Punchis district, Zamora Chinchipe province in Ecuador, and was identified by Bolivar Merino, the herbarium of Loja National University in 2012 March. A voucher specimen (accession number KRIB 0044435) of the retained material is preserved at the herbarium of KRIBB (Daejeon, Korea). 

The leaves of *G. crispiflora* were extracted with 1 L of 99.9% (*v/v*) methanol through repeat sonication and rest for 3 days at 45 °C. The resultant product was filtered with non-fluorescence cotton and concentrated using a rotary evaporator (N-1000SWD, EYELA) under reduced pressure at 45 °C, as reported previously [45]. Finally, a total of 2.15 g of methanol extract of *G. crispiflora* was obtained by freeze-drying.

### 4.3. Animals

All in vivo experiments were conducted in accordance with the guidelines of the Institutional Animal Care and Use Committee at Sungkyunkwan University (Suwon, Korea). The ICR (male, 5 weeks old) and C57BL/6 (male, 5 weeks old) mice were purchased from DAEHAN BIOLINK (Osong, Korea). All studies were conducted in agreement with the guidelines of the Institutional Animal Care and Use Committee at Sungkyunkwan University (Suwon, Korea; approval ID: SKKUIACUC2020-10-24-1).

### 4.4. Cell Culture

The investigations used the human embryonic kidney cell line HEK293T and the murine macrophage cell line RAW264.7. Each cell line was cultured in RPMI 1640 containing 1% penicillin/streptomycin (p/s) and 10% FBS at 37 °C, or DMEM containing 1% penicillin/streptomycin (p/s) and 5% FBS at 37 °C in 5% CO_2_ incubator. The RAW 264.7 and HEK293T cells were seeded into 10 cm culture plates (Falcon) at 1 × 10^6^ cells/mL and every two to three days when the confluency reached 80–90%, subcultured.

### 4.5. Nitric Oxide (NO) Assay

The RAW264.7 cells were seeded into 96-well (100 uL) plates at 1 × 10^5^ cells/mL in RPMI 1640 (supplemented with 10% FBS, 1% p/s) and cultured at 37 °C in 5% CO_2_. The cells were incubated with 50 μL of Gc-ME (50, 75, and 100 μg/mL) for 30 min. Then, the cells were incubated with 50 μL of LPS (final concentration: 1 μg/mL), poly I:C, and pam3CSK4 to measure NO production. After one day, 100 μL of supernatant was recovered and reacted with the same volume of Griess solution. Utilizing a standard curve using Bradford, the NO production was calculated from the absorbance at 540 nm. Prednisolone was used as a control anti-inflammatory drug [21] and tested under the same conditions as described above.

### 4.6. Cell Viability Assay

The RAW264.7 cells were seeded into a 96-well plate at 1 × 10^5^ cell/mL and incubated for one day. The cells were incubated with 50 μL of 50, 75, or 100 μg/mL Gc-ME or prednisolone (0–300 μM) for 24 h. After 24 h, 10 μL of MTT solution (stock concentration: 5 mg/mL) was added, and the cells were incubated for 3 h, as reported previously [46]. Then, 100 μL of supernatant was discarded, and MTT stop solution (100 μL, 10% SDS in 0.01 M HCl) was added to each well to terminate the reaction by dissolving the formazan crystal. After one day, the absorbance of the plate was measured at 540 nm.

### 4.7. Liquid Chromatography–Mass Spectrometry (LC-MS)

The LC-MS/MS analyses to determine the components of the extract were performed as described previously [47].

### 4.8. Analysis of mRNA Expression Using Reverse Transcription-Polymerase Chain Reaction (RT-PCR)

The RAW254.7 cells (2 × 10^6^ cell/mL) were seeded into a 12-well plate and pretreated with 50 or 100 μg/mL Gc-ME for 30 min. Then, LPS (final concentration 1 μg/mL) was added, and the plate was incubated for 6 h. Then, the medium was removed, and total RNA was extracted using TRizol reagent. The extracted RNA (1 μg) was used to synthesize cDNA using MuLV RT according to the manufacturer’s instructions (Thermo Fisher Scientific Waltham, MA, USA) [48]. In animal experiments, total RNA and cDNA were obtained from the sectioned stomach (LPS-induced gastritis mouse model) or sectioned lung (LPS-induced ALI mouse model). Before extraction, tissue samples were frozen using liquid nitrogen and gently homogenized to increase the surface area exposed to the TRizol™. Synthesized cDNAs were used for RT-PCR, and the DNA products were loaded into 1% agarose gel containing EtBr to confirm the band intensity. Table 1 lists the primer sequences exploited for RT-PCR.

### 4.9. Luciferase Reporter Gene Activity Assay

HEK293T cells were seeded into 24-well plates at a concentration of 2 × 10^5^ cells/mL (DMEM) and incubated for 24 h. The cells were co-transfected with either 0.44 μg of MyD88 or TRIF adaptor molecules, 0.44 μg of the NF-κB-Luc (luciferase) gene or AP-1-Luc gene, and 0.11 μg β-galactosidase using PEI. After 24 h, Gc-ME (50 and 100 μg/mL) was added, and the cells were incubated for another 24 h. Luciferase activity was measured with a luminometer using luciferin bioluminescence, and the results were normalized to β-galactosidase activity [49].

### 4.10. Western Blotting Analyses

The RAW264.7 cells were cultured in a six-well plate at 5 × 10^6^ cell/mL and incubated for 24 h. Then, Gc-ME (100 μg/mL) was added for 30 min, followed by LPS (final concentration 1 μg/mL). The cells were incubated for 5, 15, 30, or 60 min for p50, p65, IκBα, and AKT assays and for 2, 3, or 5 min for Src. At specific time points, the cells were collected in cold PBS and ruptured with cell lysis buffer. After centrifugation at 12,000 rpm for 1 min, the supernatant (protein) was collected, and the cell debris was discarded. Homogenization buffer A (20 mM Tris-HCl (pH 8.0), 10 mM EGTA, 2 mM EDTA, 2 mM DTT, 1 mM PMSF, 25 μg/mL aprotinin, and 10 μg/mL leupeptin) was used to obtain nuclear lysates. After sonicating and centrifuging, homogenization buffer B (buffer A, 1% Triton X-100, Sigma, St. Louis, USA) was used for resolving the cells. In the Src-domain deletion assay, HEK293T cells transfected for 24 h were collected and ruptured with cell lysis buffer. Animal tissues (sectioned stomachs and lungs) from the gastritis and ALI model mice were also homogenized after freezing using liquid nitrogen and ruptured using cell lysis buffer. Protein samples from total cell lysate, nuclear lysate, and animal tissue (stomach lysate from gastritis model and lung lysate from ALI model) were loaded onto SDS-polyacrylamide gel and electrophoresed (Bio-Rad, Hercules, CA, USA) [13,50]. Western blotting analyses were executed with specific primary and secondary antibodies for each target protein, and ECL solution was used to detect the antibody.

### 4.11. In Vivo HCl/EtOH-Induced Acute Gastritis Mouse Model

The ICR mice (*n* = 3 to 5 per group, male, 5 weeks old) were housed in plastic cages for five days for acclimatization. All drugs used were dissolved in 0.5% CMC solution. In order to induce acute gastritis in the mouse model, HCl (150 mM)/EtOH (60%) was used. After fasting for 24 h, ICR mice were administered with Gc-ME (50 or 100 mg/kg), ranitidine (40 mg/kg), or 0.5% CMC by oral administration three times at 12 h intervals [41]. Acute gastritis was induced by administering 300 μL of HCl (150 mM)/EtOH (60%) with a gavage eight hours after the last injection. Mice were given isoflurane anesthesia and then sacrificed an hour later. Gastritis lesions were recorded photographically, and the stomach was washed in PBS and used for mRNA and protein extraction.

### 4.12. In Vivo LPS-Induced ALI Model

All drugs were dissolved in PBS. The C57BL/6 mice (n = 5 per group, male, 5 weeks old) were administered Gc-ME (50 and 100 mg/kg), dexamethasone (5 mg/kg), or PBS twice at 6 h intervals before lipopolysaccharide (LPS) induction. One hour after the second administration, 50 μL of LPS (5 mg/kg) was administered intranasally in both nostrils twice at 1 h intervals, except for the control mouse group that received intranasal PBS. A third oral administration of Gc-ME or dexamethasone was offered following induction. The mice were anesthetized and sacrificed by CO_2_ after a 16 h fast, and the lungs were taken. The left lung was washed in PBS, and the wet lung weight was measured.

After drying the lung in an oven at 75 °C for 48 h, the weight of the dried lung was measured, and the lung wet-weight:dry-weight ratio was calculated as a measure of the degree of pulmonary edema [41,49]. To evaluate the histological changes, the upper-right part of the lung was excised, fixed in 4% paraformaldehyde for 24 h, and then embedded in paraffin solution before being cut into 4 μm-thick tissue sections. Sectioned lung tissues were stained with hematoxylin-eosin (H&E) and observed by light microscopy. According to the acute lung injury scoring system shown in Table 2, the number of neutrophils, the number of hyaline membranes, alveolar septal thickening, and the amount of proteinaceous debris in the airspaces were recorded. The lower lobe of the right lung was frozen in liquid nitrogen, homogenized, and used for mRNA (RT-PCR) or protein (Western blot assay) extraction and analyses.

### 4.13. Cellular Thermal Shift Assay (CETSA)

HEK293 cells treated DMSO (for control) or Gc-ME (100 μg/mL) were collected and heated at a temperature gradient from 42–65 °C, and the following steps including immunoblotting analysis were performed as previously reported [51].

### 4.14. Statistical Analyses

All data used in this study are presented as mean ± standard deviation (SD) calculated from at least six replicates for in vitro experiments and eight mice per group for in vivo studies. Using Student’s *t*-test and the Mann–Whitney test, the statistical significance of the difference between the results for the various experimental groups and the control group was determined. A *p*-value of 0.05 or lower was deemed statistically significant.

## 5. Conclusions

The findings of this study suggest that Gc-ME mediates anti-inflammatory effects by targeting Src kinase in macrophages activated by various pathogenic stimuli such as LPS or damage inducers such as HCl/ethanol treatment as summarized in Figure 6. In both in vitro and in vivo models, Gc-ME reduced NO production, the transcriptional expression of inflammation-related genes, and the phosphorylation of NF-κB signaling proteins. These activities were found to be the result of Gc-ME binding to the kinase domain of Src, the most upstream signaling molecule of the NF-κB pathway. In animal models, Gc-ME also showed anti-inflammatory activity to ameliorate acute gastritis induced by HCl/EtOH and acute lung injury induced by LPS. These results suggest Gc-ME as a potential anti-inflammatory herbal drug to treat inflammatory diseases including gastritis and acute lung injury.

## Figures and Tables

**Figure 1 plants-11-03560-f001:**
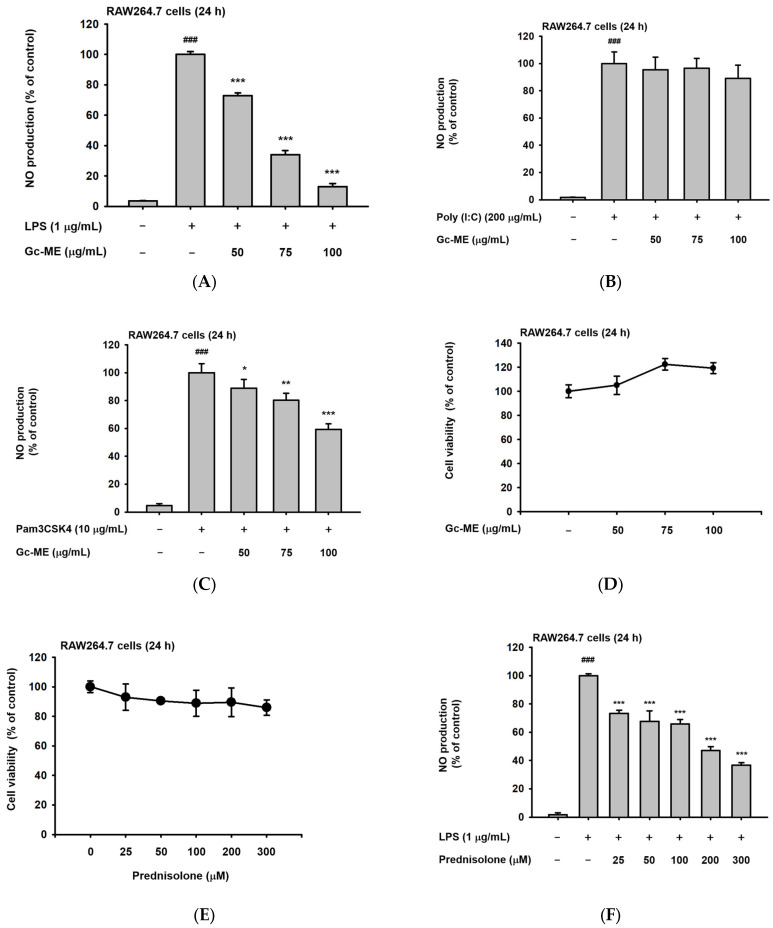
Effects of *Guettarda crispiflora* Vahl methanol extract (Gc-ME) on NO production and cytotoxicity in RAW264.7 cells. (**A**–**C**) Gc-ME reduces lipopolysaccharide (LPS; 1 μg/mL)-induced NO production (upper panel), polyinosinic: polycytidylic acid (Poly (I: C)) (200 μg/mL) (middle panel), or Pam3CysSerLys4 (Pam3CSK4) (10 μg/mL) (lower panel) induction in a concentration-dependent manner (50–100 μg/mL). (**D**) Gc-ME (50–150 μg/mL) did produce cytotoxicity in RAW264.7 cells. (**E**) Effect of prednisolone (0–300 μM) on the viability of RAW264.7 cells was examined by MTT assay. (**F**) Reduction in NO production when incubated with prednisolone at various concentrations (0–300 μM). (**G**) Types of phytochemicals identified in the Gc-ME preparation by LC/MS-MS. Results (**A**–**F**) are expressed as mean ± standard deviation. ###, *p* < 0.01 compared to control group (no treatment); *, *p* < 0.05; **, *p* < 0.01; ***, *p* < 0.001 compared to control group (LPS alone) by Student’s t-test; −, no treatment; +, treatment. Red circle: Retention time of putatively active compounds.

**Figure 2 plants-11-03560-f002:**
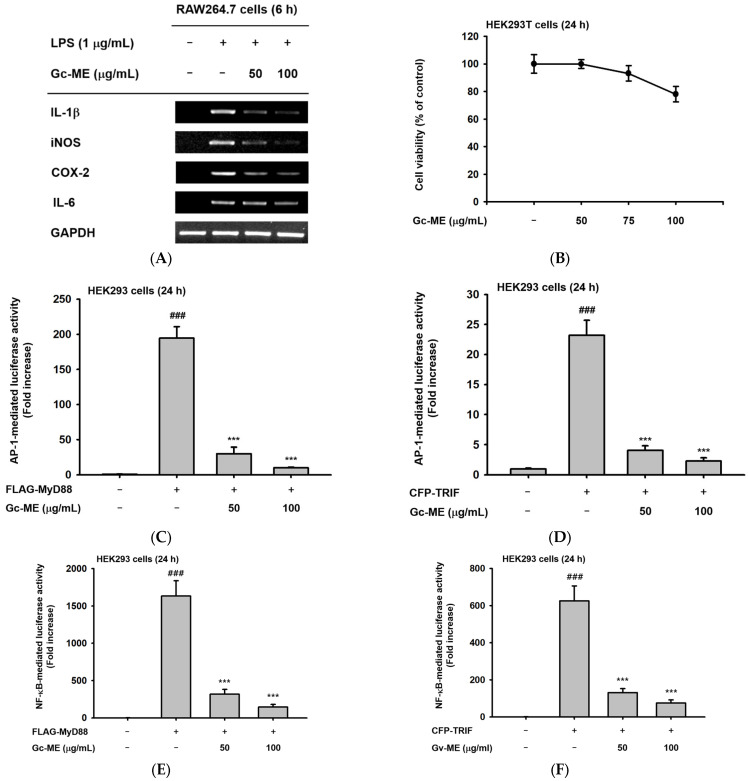
Effects of Gc-ME on the expression of pro-inflammatory cytokine mRNA and transcription factors. (**A**) Pretreatment with Gc-ME (50–100 μg/mL) reduced inflammatory cytokine mRNA expression in LPS-stimulated RAW264.7 cells. (**B**) Gc-ME was not cytotoxic in HEK293T cells at 50–100 μg/mL. (**C**–**F**) Gc-ME (50–100 μg/mL) reduced the activation of the AP-1 and NF-κB transcription factors. HEK293T cells were co-transfected with a luciferase reporter vector containing a promoter part of each transcription factor and a β-gal plasmid control with or without MyD88 (left panel) or TRIF (right panel). (**G**) Total and phosphorylated p50, p65, and β-actin were analyzed using an immunoblotting assay. The RAW264.7 cells were pretreated with Gc-ME (100 μg/mL) for 30 min, and the inflammatory responses induced by LPS (1 μg/mL) were analyzed by induction time. Results (**B**–**F**) are expressed as mean ± standard deviation. ###: *p* < 0.01 compared to normal group (no treatment), ***: *p* < 0.001 compared to control group by Student’s *t*-test. −: no treatment, +: treatment.

**Figure 3 plants-11-03560-f003:**
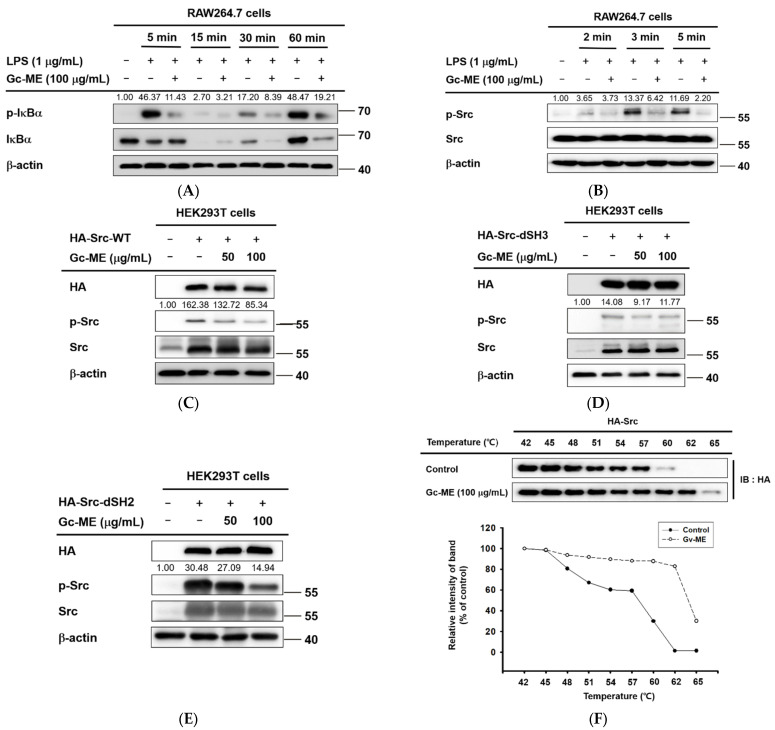
Effects of Gc-ME on Src phosphorylation. (**A**,**B**) RAW264.7 cells were incubated with Gc-ME (100 μg/mL) for 30 min, and the inflammatory response was induced by adding LPS (1 μg/mL) to each group for a fixed period of time. Phosphorylated and total forms of IκBα and Src were detected by Western blotting. (**C**–**E**) Identification of domains to which Gc-ME binds. HEK293T cells were transfected with HA-tagged wild-type Src or mutant Src vectors and incubated with Gc-ME (50–100μg/mL). Phosphorylated and total forms of Src, HA, and β-actin were detected by Western blotting. (**F**) The stabilizing effects of Gc-ME interaction with the Src protein were confirmed using the cellular thermal shift assay (CETSA). Western blotting was performed on the cell lysates, and Src band intensity was assessed using Image J software. −: no treatment, +: treatment.

**Figure 4 plants-11-03560-f004:**
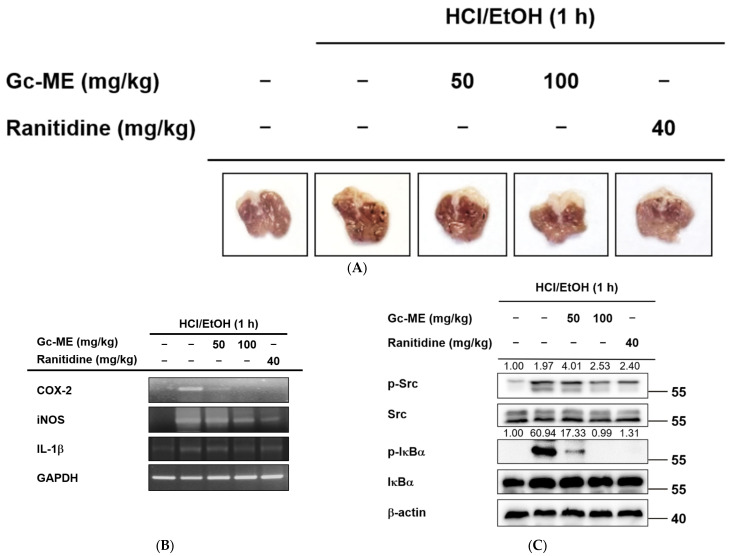
Gc-ME reduced gastritis lesions and inflammatory signaling induced by HCl/EtOH. (**A**) Mice were starved for one day before Gc-ME administration. The ICR mice (n = 3 to 5 per group, male, 5 weeks old) were administered Gc-ME (50–100 mg/kg) or ranitidine (40 mg/kg) orally three times at eight h intervals by a gavage. The control group was administered 0.5% CMC by oral injection. HCl/EtOH (150 mM) treatment proceeded after one hour, and then the mice were sacrificed. The stomachs were excised, and images of the inner surface of the stomach were obtained. (**B**,**C**) Sectioned stomachs from 3 to 5 mice were frozen in liquid nitrogen, homogenized, and underwent extraction for preparing total mRNA and protein. The COX-2, iNOS, IL-1β, and IL-6 mRNA levels were measured by qRT-PCR. The p-Src and p-IκBα levels were detected by Western blotting. −: no treatment, +: treatment.

**Figure 5 plants-11-03560-f005:**
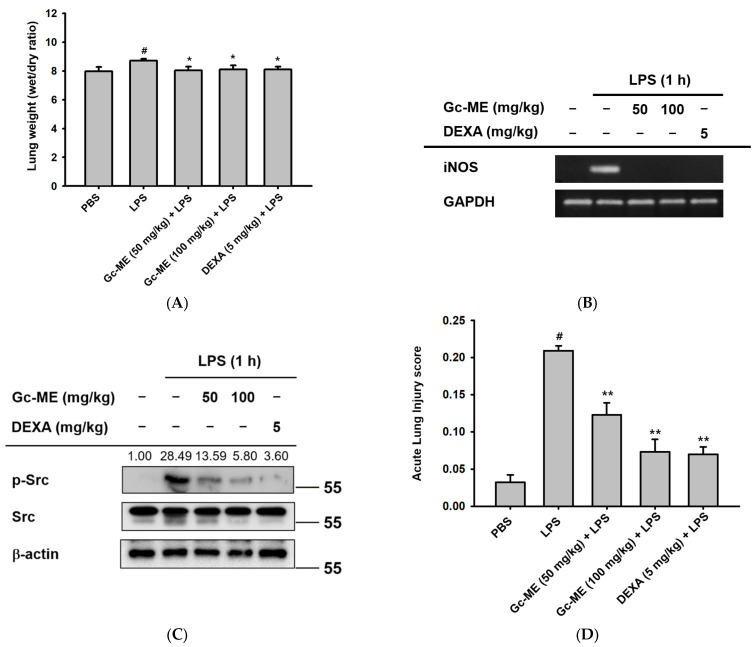
Gc-ME alleviated histopathological changes in acute lung injury (ALI) in an LPS-induced mouse model of ALI. (**A**) Mice were orally administered Gc-ME (50–100 mg/kg) or dexamethasone (5 mg/kg) or PBS twice at six h intervals by gavage. One hour after the last oral administration, ALI was induced by the intranasal administration of LPS (5 mg/kg) two times at one h intervals. One hour after LPS induction, Gc-ME, dexamethasone, or PBS was administered one more time. The lung wet-to-dry weight ratios for each group were measured 16 h after ALI induction. (**B**,**C**) After 16 h of induction, the sectioned lungs from 5 mice were frozen in liquid nitrogen and homogenized and mRNA and protein were extracted. iNOS mRNA expression was measured using qRT-PCR. (**D**,**E**) The severity of acute lung injury was assessed microscopically (at 4×, 10×, and 20× magnification) using H&E-stained lung sections in accordance with the American Thoracic Society lung injury scoring system. All data (**A**–**D**) are presented as mean ± standard deviation (SD) calculated from at least five mice or samples. #: *p* < 0.01 compared to normal group (no treatment), *: *p* < 0.05 and **: *p* < 0.01 compared with the non-induced group. –: no treatment, +: treatment.

**Figure 6 plants-11-03560-f006:**
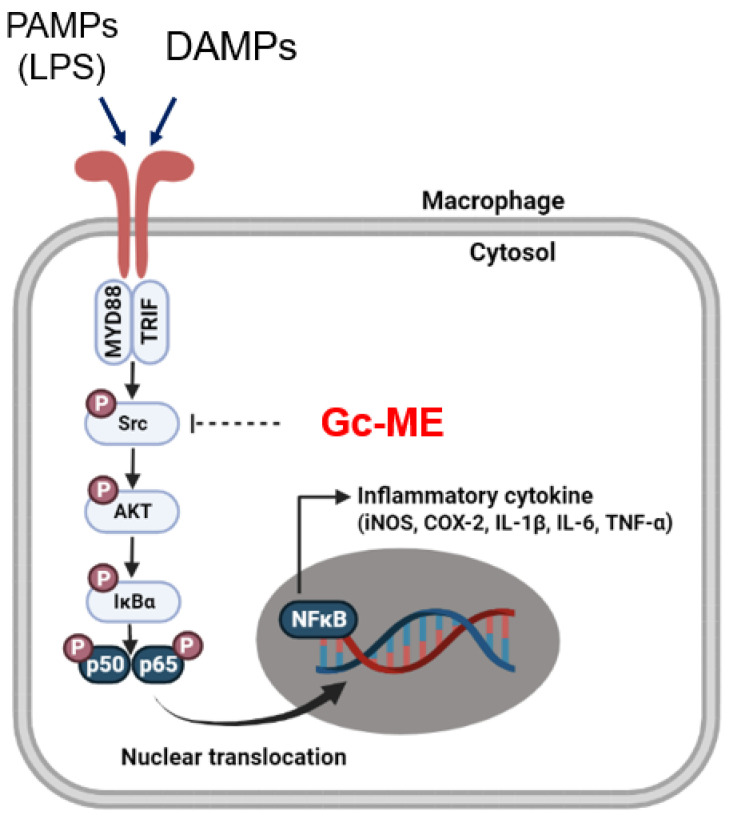
Molecular mechanism of Gc-ME indicating anti-inflammatory activity targeting Src.

**Table 1 plants-11-03560-t001:** RT PCR primer sequences.

Gene (Type)	Direction	Sequences (5′ to 3′)
COX-2	Forward	TCACGTGGAGTCCGCTTTAC
	Reverse	TTCGACAGGAAGGGGATGTT
iNOS	Forward	TGCCAGGGTCACAACTTTACA
	Reverse	ACCCCAAGCAAGACTTGGAC
IL-1β	Forward	CAGGATGAGGACATGAGCACC
	Reverse	CTCTGCAGACTCAAACTCCAC
IL-6	Forward	GCCTTCTTGGGACTGATGG
	Reverse	TGGAAATTGGGGTAGGAAGGAC
GAPDH	Forward	GAAGGTCGGTGTGAACGGAT
	Reverse	AGTGATGGCATGGACTGTGG

**Table 2 plants-11-03560-t002:** Acute lung injury scoring.

Parameters	Scoring		
	0	1	2
(A) Neutrophils in the alveolar space	None	1 to 5	>5
(B) Neutrophils in the interstitial space	None	1 to 5	>5
(C) Number of hyalin membrane	None	1	>1
(D) Amount of proteinaceous debris filling in the airspace	None	1	>1
(E) Alveolar septal thickening	<2×	2×–4×	>4×
Scoring = [(20 × A) + (14 × B) + (7 × C) + (2 × D)]/(number of fields × 100)			

## Data Availability

Not applicable.

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
