# Peer review of "Guettarda crispiflora Vahl Methanol Extract Ameliorates Acute Lung Injury and Gastritis by Suppressing Src Phosphorylation"

_plants, 2022, doi:10.3390/plants11243560_

Round 1
Reviewer 1 Report
Comments to the Authors
Manuscript by Dahee Lee …….. and Jae Youl Cho presents an interesting concept of using Guettarda crispiflora Vahl methanol extract to ameliorate ALI and gastritis by targeting Src/NF-kB pathway. These kinds of studies are helpful in determining if the natural products can have positive impact on human health. Title needs to be modified as the authors have not addressed the role of NF-kB under in vivo conditions. Although the authors have carefully conducted the study but there are still some major concerns listed as under that needs to be addressed:
11. Quantitative RT-PCR is the norm in all the manuscripts now-a-days, therefore, in Figure 2A and in all the subsequent figure, authors should perform qRT-PCR and should give the quantitative data for all the cytokines studied.
22. Please provide the molecular weights of proteins in the western blots so as to benefit readers.
3. For the benefit of readers, quantitation for the western blots in Figure 2E should be performed and data for phosphorylated proteins should be normalized to the respective total protein within the same signaling molecule.
44. Similarly, for the benefit of readers, quantitation for all the western blots in Figure 3 should also be performed and data for phosphorylated proteins should be normalized to the respective total protein within the same signaling molecule.
55. There is some discrepancy in Figure 3A regarding the IkBa protein. p-IkBa protein at 15 minutes and IkBa protein at 30 minutes does not look right. Authors have also provided no explanation in this regard in the manuscript. Please comment on these time points in the western blots of Figure 3A.
66. Authors have reported that they have used 8 mice/group in Figure 4. However, it is completely unclear from the manuscript that what was the criterion for them to choose the mice as the authors are just showing one image each in Figure 4A. Is this the representative image. Authors are encouraged to show at least three images each of the stomach and include it as supplementary data.
77. Similarly, regarding the Figure 4B and 4C, authors are showing the mRNA expression and western blots from one of the mice from 8 mice. What was the criterion for choosing just one mouse for the analysis. This should be explained in detail in the manuscript so that readers can understand the rationale for using just one mice from the group of 8 mice/group.
88. qRT-PCR should be performed in Figure 4A and quantitation for western blot analysis should be performed as pointed out in point 3.
99. qRT-PCR should also be performed in Figure 5B and quantitation for western blot analysis in Figure 5C should be performed as pointed out in point 3.
10. Involvement of NF-kB in Figure 6 is just a speculation, as no data has been shown under in-vivo conditions. Authors should modify this figure accordingly.
Reviewer 2 Report
In this paper, it is demonstrated Guettarda crispiflora Vahl methanol extract ameliorates acute lung injury and gastritis by targeting the Src/NF-κB pathway. The structure of the paper is reasonable and the content is comprehensive. However, there are still some problems and suggestions for further revision and improvement:
1. It is suggested to supply how LPS, poly(I:C) or Pam3CSK4 induce inflammation.
2. The description of experimental results is inconsistent with the data graph.
3. It is suggested to increase the statistical chart of western blotting results.
4. It is recommended to give clear pictures.
5. The experimental results did not mark the position of the picture in the part of 2.5.
6. It is suggested to add relevant references in the part of the discussion.
7. The inhibition of NO production by Gc-ME was not statistically significant when the RAW264.7 cells were co-incubated with Poly (I: C), which is another stimulus NO production tended to decrease with concentration and showed a significant effect on TLR2 activation by Pam3CSK4 (Figure 1A, middle-lower panel). In this word, how does Pam3CSK4 activate TLR2 to have a significant impact?
8. The units in the figures are not uniform, for example: Figure 1D.
9. The statement earlier in the article that Gc-ME does not produce cytotoxicity to RAW264.7 or HEK293T cells under any concentration test (Figure 1B) contradicts the statement that (B) Gc-ME (50-150 μg/mL) does produce cytotoxicity to RAW264.7 cells in Figure 1 later.
10. Poly(I:C) is not statistically significant, why choose such a NO stimulus and use the data in the article.
11. First, a representative pro-inflammatory cytokine (eg., IL-1) was selected, and the expression of the corresponding mRNA was confirmed through RT-PCR. Please add the corresponding expression result diagram.
12. The 15-minute result in Figure 3 is incorrect.
13. At all induction time points, phosphorylation of each factor is reduced (Figure 3A). Please add other factor plots. For example, factors P50, P65.
14. There is skepticism about the statistical significance of Figure A-plot in Figure 5.
